# The Effects of Performance of Public Sector Health System on Quality of Life in China: Evidence from the CGSS2015

**DOI:** 10.3390/ijerph17082896

**Published:** 2020-04-22

**Authors:** Zongfeng Sun, Jintao Li

**Affiliations:** 1School of Political Science and Public Administration, Shandong University, Qingdao 266237, China; sunzongfeng2017@sdu.edu.cn; 2Institute of Governance, Shandong University, Qingdao 266237, China

**Keywords:** public sector health system, QOL, ordered logit model, mixed effects order logit model

## Abstract

The determinants of quality of life can be social, economic, cultural, and governmental, to name a few. Prior research has shown that demographic factors and social support can have a significant impact on an individual’s quality of life. This research attempts to examine the effects of public sector health system performance on individuals’ quality of life in China, measured by (1) self-reported health, (2) depressed mood, and (3) happiness. The targeted population was individuals aged 18 and above. The primary data was collected from the CGSS2015 (Chinese General Social Survey 2015), and the National Statistical Yearbook was also used. Using various statistical models, this study finds that the correlation coefficients of government performance in providing healthcare for patients on quality of life are 0.144, 0.167, and 0.328, respectively. The more satisfied with medical care and public health delivered by the government, the higher the level of quality of life. In addition, the relationship between government expenditure on public health service and quality of life is positively significant. These findings are robust after weighting methods are used. The performance of the Public Sector Health System has a significantly positive association with quality of life in China.

## 1. Introduction

Over the past several years, researchers have investigated quality of life (QOL) and the general well-being of individuals, outlining both the positive and negative features of their lives. The extant literature exploring the driving forces of QOL can be divided into three approaches. The first is rooted in demographic factors such as being female, being older, living alone, living in disadvantaged conditions, having a lower socioeconomic status, being injured, and experiencing deaths of loved ones during emergencies, which have all been shown to significantly decrease the level of one’s QOL [1,2,3,4]. The second approach to studying the driving forces of QOL, focuses on the effects of events such as disasters [5], hurricanes [6], and terrorism [7] on QOL. The third approach investigates how social support can enhance the QOL among different groups [8,9]. However, little research has paid attention to the role of government delivery of public health services in advancing public QOL.

In recent years, more and more attention has been paid by researchers to the effect the government has on the public’s life satisfaction [10,11,12]. There are many ways in which the government can play an important role in the public’s daily life, such as in policy making, policy implementation, and policy outcomes. Among these, the most critical should be policy outcomes, which can be measured by public service performance. Accordingly, public service performance can impact citizens’ trust in the government [13,14]. After four decades of development in public health and reform in healthcare and medicine, significant progress has been made in developing a system of public health service. For example, a basic medical service network covering both urban and rural areas has been instituted, with 980,000 medical and health institutions at all levels, seven million beds within those institutions, and 1.17 million health workers [15] by 2017. The Chinese government gives priority to prevention in combination with treatment and makes great efforts to ensure equal access to the public health service. It also devotes much energy to preventing and controlling epidemics, chronic and endemic diseases, strengthening the capacity for quick responses to public health emergencies, and developing an increasingly equal and universal basic public health service system [16]. However, the effect of the government’s public sector health service delivery on public QOL in China has been rarely studied in prior research. Although the association between government performance and life satisfaction has been studied in Britain [17], less research has been carried out in developing countries, especially in China. 

Therefore, the goal of this study was to fill this research gap by investigating the effects of performance of the public sector health system on QOL in China. We employed China’s General Social Survey (GSS) 2015 and the National Statistical Yearbook 2015 as the data sources for this article. Using an objective and subjective measurement of the variables, this study attempts to examine the relationship between performance of public sector health system and QOL, and provides a possible mechanism for their relationship. The possible contribution of this paper is threefold. Firstly, the study is an attempt to evaluate the performance of the public sector health system in China from the perspective of the public. Secondly, the association between the performance of the public sector health system and QOL is investigated using various statistical methods, where little research has been done in prior studies. Thirdly, an explanation is offered based on the theoretical model of IPOO in Public Administration. 

Different from the concept of mere life satisfaction, QOL observes life satisfaction as well as everything from physical health, family, education, employment, wealth, safety, the security of freedom, religious beliefs, and the environment. Standard indicators of QOL include not only wealth and employment but also the built environment, physical and mental health, education, recreation and leisure time, and social belonging [18]. According to the World Health Organization (WHO), QOL is defined as “the individual’s perception of their position in life in the context of the culture and value systems in which they live and in relation to their goals.” In comparison to the WHO’s definition, the Wong-Baker Faces Pain Rating Scale defines QOL as “life quality (in this case, physical pain) at a precise moment in time” [19]. Within the field of healthcare, QOL is often regarded in terms of how a certain ailment affects a patient on an individual level. Therefore, the definition of QOL varies across different disciplines. 

Consequently, the methods for the measurement of QOL can be varied. The per capita gross domestic product (GDP) or standard of living can be used to measure QOL in terms of financial aspects, while, emotional well-being—measuring respondents’ emotional experiences—is a second approach that is often applied. In addition, there is a third method, considering happiness, the subjective state of mind, as QOL. In practice, there are some famous agencies and NGOs publishing various indexes to reflect the QOL among different states, such as the World Happiness Report, the Human Development Index, the Physical Quality of Life Index, the Where-to-be-born Index, and so on. As suggested at the very beginning of this article, there are many factors that might impact the variation of QOL, such as demographic factors, events, and social support. However, little research has been done on the public sector health system.

The development of public health in China has changed dramatically since its reform and opening-up in 1978 and, soon after, a social medical insurance system to cover the basic medical needs of workers began to form. Then, in 2000, the goal of establishing an urban medical and healthcare system, in line with the socialist market economy, was set up so that the people could enjoy a reasonably priced, high-quality medical service, and thus lead healthier lives. Furthermore, in 2002, the government decided to drive health service reform to a deeper level, and put more funding into rural areas, to provide different levels of medical services to rural residents. In fact, there are three kinds of medical services covering the rural area in China, delivered by county hospital, town health center, and village clinic. Correspondingly, the levels of medical service vary among these different agencies. The county hospital delivers the highest quality of medical services among them, while the village clinic provides the lowest quality of medical services. These levels of medical services are open to all the rural residents instead of those who live in the urban area. In 2009, a new round of reform of the medical healthcare system has been initiated by China’s government, with a goal to provide access to the basic medical and healthcare system for all citizens. Even in 2012, the reform and public health service were developed further. For example, the reform of public hospitals and the price reform of drugs and medical services have been enhanced and serious illness insurance policies covering both urban and rural residents have been implemented. 

Although significant progress has been made in developing a system of medical and health services, many problems still exist, such as inequality in the public health service. Studies show that residents in the developed provinces of China have better public health services than those in undeveloped provinces, which demonstrates that the inequality in the public health service can be a serious problem [20]. For example, infant mortality rates and mortality rates of children under five years old in the undeveloped western region of China in 2005 was more than three times those in the developed, eastern region [21]. The number of medical clinic visits per person in developed provinces such as Beijing and Shanghai in 2014 was 9.93 and 10.33, respectively, while that number in undeveloped provinces such as Guizhou and Qinghai was only 3.71 and 3.87 [22]. Therefore, many studies have attempted to establish the reasons why the inequality of the public health service has been so serious in China. It is obvious that the causes of inequality can be complex and vary not only by region but also over time. Extant research, which explain the inequality in the public health service, can be divided into various approaches such as socioeconomic factors [23], policy implementation and health system reform [24], China’s unique political system which prioritizes economic development [20], and the transportation barriers to accessing healthcare services in rural areas [25,26].

Despite those factors which drive inequality in the public health service in China, research should pay more attention to the results of this inequality. One result, for instance, is that inequality might bring lots of complaints and a sense of unfairness once the inequality has spread, and this will unavoidably affect everyone [27]. Although the relationship between health at an individual level and life satisfaction has been investigated in previous research [28], the association between the performance of the public sector health system and QOL in China has been rarely examined. 

According to the Input-Process-Output-Outcome (IPOO) model [29], the process for public health administration can be divided into four stages: the input, the process, the output, and the outcome, as shown in Figure 1. In the input stage, many health resources, such as medical and health institutions, health workers, and beds at medical institutions are provided by governments and other organizations. The process stage, also called activities, specifically refers to public health management, such as the reform of agencies, the formulation of new institutions, the promotion of national health education, and so on. The output stage reflects the results of the input and process stages of public health, which can be important for public health evaluation within governments. Finally, the outcome stage reflects the customers’ evaluation, such as whether they are satisfied with the public health services delivered. For example, the indicators of satisfaction with public health can be self-reported health, and the QOL (dependent variable in this article). According to the IPOO model, the output indicators (public health service) should influence the outcome, such as QOL, which is also the basic idea behind New Public Management.

Many studies have been conducted to prove the equivalence of the output and outcome indicators and that they can be used similarly and should have equal attention paid to them [30,31]. However, there is some evidence against this conception of the relationship between the outcome and the output variables in different fields, such as in police performance, public utilities, and so on. For instance, citizens’ satisfaction with public services is not always congruent with the objective of those services [32,33]. Other researchers argue that the level of expectation between the output and outcome should be of substantial importance [34].

The output of a public health service, which can be directly linked to public daily life, can influence their QOL through the performance legitimacy theory [35,36]. No matter under what regime a country operates, sustainable development matters, and can be realized through good public sector services. Prior research has shown that higher levels of public services delivered by governments resulted in higher levels of citizen trust in government [14,37], which makes citizens perceive that a society is more livable [38], therefore more happiness will be generated on average. In addition, evidence from Korea in 2013 showed that trust in government affected happiness both directly and indirectly [39]. In sum, there is a significantly positive relationship between public service performance and citizens’ quality of life in terms of happiness. 

As an important category of public service, a good public health service delivered by a public sector system can enhance citizens’ trust in governments and society, which makes them believe that the society with a shared higher public health level is more livable. Patients believe that the possibility that disease can be cured will be higher in places with a good public health service performance than in poorer areas. As such, their hope for high QOL can be ignited by a higher level of public health with the idea that diseases can be overcome. In addition, the good performance of a public sector health system ensures that laws and regulations that protect health and safety are enforced. Necessary healthcare and health services should be provided in such a way that the public gets an accessible, effective, and high-quality public health service. Healthcare programs in urban and rural areas decrease the costs of visiting doctors; that is, the odds ratio of serious illness causes a decline in poverty. 

Therefore, we hypothesized that the performance of public sector health systems can significantly enhance QOL in China.

## 2. Materials and Methods 

### 2.1. Data Collection

The data used in this paper is drawn from the 2015 Chinese General Social Survey (CGSS 2015), which is a representative annual survey in China, and it was conducted by the National Survey Research Center of China. The CGSS 2015 covered 478 villages from 28 provinces, cities, and autonomous regions across China, and employed a stratified multistage probability proportional to size sampling design. The CGSS 2015 has a valid response sample size of 10,968 and the respondents of the survey are the members of the whole family. The population consists of people 18 years of age or older. The dataset’s main variables include respondents’ employment, family, households, life satisfaction, QOL etc. Data is available through the Chinese National Survey Data Archive Website. In addition, we collected data such as public expenditure on public health, GDP (Gross Domestic Product) per capita, and population from the China Statistical Yearbook 2015 to measure the objective public sector service performance with expenditure on health.

### 2.2. Variables Specification

Quality of life: In line with previous studies [4,40], QOL was captured by three variables including perceived happiness (“Overall, how satisfied are you with your current life?”), self-reported health (“Overall, how do you feel about your health?”), and depressed mood (“Overall, how often do you feel that you are in depressed mood?”). Both perceived happiness and self-reported health were measured by a five-point Likert table, representing the increased degree of each indicator. Conversely, as for the variable of depressed mood, the bigger number represents the better moods. In order to examine the reliability of the three items as one measurement component, we use Cronbach’s alpha test, and its scale reliability coefficient is 0.8532, which is good according to Kline [41].

The Performance of Public Sector Health System. Drawing from Zineldin [42] and Hussain [43], we used both objective and subjective methods to measure the independent variable in this paper. The subjective measure consists of two general questions: (1) “Overall, to what extent are you satisfied with the governments’ performance in providing health care for patients?” The question can reflect the citizen’s subjective evaluation of government performance in providing healthcare for patients, which is short for HCP (Healthcare for patients). The variable was measured by a five-point Likert table, representing the increased degree with a bigger number; (2) “Overall, to what extent are you satisfied with the medical care and public health provided by the government?” The question here stands for the citizens’ satisfaction with medical care and public health, short for MCPH (medical care and public health). The variable is continuous, ranging from 0 to 100, with 100 representing “very satisfied”. The objective measure was based on public expenditure on health. Theoretically, the more expenditure on public health, the higher the level of performance of the public sector health system.

Control variables: Primarily, demographic variables such as gender, age, age squared, education attainment, party status (for example, Communist Party of China member (CPC member), Democratic party (Chinese eight democratic parties, namely Revolutionary Committee Of The Chinese Kuomintang, China Democratic League, China National Democratic Construction Association, China Association for Promoting Democracy, Chinese Peasants and Workers Democratic Party, China Zhi Gong Party/China Public Interest Party, Jiu San Society, and Taiwan Democratic Self-Government League), Youth league member, and the Mass.), place of residence (urban and rural areas), marital status (married, divorced, widowed, and non-married), and annual income were included as controlled variables. Information sources such as internet usage have proven to have a significant influence on QOL [44]. In the CGSS 2015, internet usage was measured by asking the respondent directly: “In the past one year, how often have you usually used the internet?” The answer ranged from 1 to 5, representing the increased degree of each indicator. Frequencies and percentages for categorical variables, means, and standard deviations for continuous variables for the entire sample are reported in Table 1.

### 2.3. Model Specification

The dependent variable was specified as three questions: self-reported health, depressed mood, and happiness. Those three questions were rated through the Likert tables ranging from 1 to 5, to which an ordered logit model should have been applied. However, this sampling is designed by stratified multistage probability, proportional to size, which means that a weighted regression model should be also considered when running regression models as shown in Equation (1). The regression results shown in Table 2 reflect the actual weighed ordered logit model. In addition, when we use objective statistical data to measure the performance of public sector health system, two levels of data structure coexist in one model. Therefore, a hierarchical model should also be used to obtain a robust parameter estimation, demonstrated in Equation (2). Table 3 shows the hierarchical model estimated for the effects of health performance.
(1)$$\begin{array}{l} \log it(1)=\ln(\frac{P_{2}{+P}_{3}{+P}_{4}{+P}_{5}}{P_{1}})=\alpha 1+\sum \beta_{i}\chi_{i}\\ \log it(2)=\ln(\frac{P_{3}{+P}_{4}{+P}_{5}}{P_{1}{+P}_{2}})=\alpha 2+\sum \beta_{i}\chi_{i}\\ \log it(3)=\ln(\frac{P_{4}{+P}_{5}}{P_{1}+P_{2}+P_{3}})=\alpha 3+\sum \beta_{i}\chi_{i}\\ \log it(4)=\ln(\frac{P_{5}}{P_{1}+P_{2}{+P}_{3}{+P}_{4}})=\alpha 4+\sum \beta_{i}\chi_{i} \end{array}$$

The ordered logit model simultaneously estimates multiple equations. The number of equations will be the number of categories in the dependent variables minus one. All three dependent variables are 5-point scale; therefore, 4 equations have to be specified. Each equation models the odds ratio of the set of categories on the above score line to the set of categories below. In the above equations, the *α*1, *α*2, *α*3, and *α*4 are the intercepts to be estimated, and the ∑*β_i_χ_i_* denotes the independent variable *i* and coefficient *i*.

The two-level model was also considered, where, for a series of M independent clusters and conditional on a set of fixed effects *X_ij_*, a set of cut-points κ, and a set of random effects μ_j_, the cumulative probability of the response being in a category higher than κ could be expressed as follows:(2)$$\mathrm{Pr}(y_{ij}>k|X_{ij},\kappa ,\mu_{j})=H(X_{ij}\beta +Z_{ij}\mu_{j}-\kappa_{k})$$
for j = 1,…,28, M clusters, with cluster j consisting of *i* = 1,…,$n_{j}$ observations. The cut-points κ are labeled K_1_, K_2_,…, K_K−1_ where *K* is the number of possible outcomes. *H*(·) is the logistic cumulative distribution function that represents cumulative probability. 

## 3. Results

### 3.1. Description of Quality of Life in China

We first report the QOL in China in 2015, as shown in the Figure 2, Figure 3 and Figure 4.

As Figure 2 shows, there were 37.83% of respondents who felt very healthy and 20.18% in complete health, whereas, 22.61% of respondents rated themselves as somewhat healthy. There were 15.87% respondents who did not think that they were very healthy and 3.51% who did not feel healthy at all.

Figure 3 shows the distribution of respondents who experienced depressed moods. Accordingly, 24.59% of respondents indicated that they had never experienced a depressed mood and the proportion of respondents indicating seldom was 42.05%. There were 24.51% of respondents who sometimes experienced a depressed mood. Comparatively, there were 7.52% of respondents who expressed that they often felt depressed. Furthermore, there were 1.32% of respondents who always felt depressed. 

According to Figure 4, 58.49% of respondents indicated that they were very happy and 17.31% experienced complete happiness. There was a 15.48% proportion of respondents who were somewhat happy. Comparatively, the number of respondents who were not very happy was 7.05% and a total of 1.67% were not happy at all.

### 3.2. Regression Results

According to Table 2, there is a positive relationship between performance in providing healthcare for patients and QOL, while controlling for socio-demographic and social capital variables, which is significant at 0.001 level. To be more specific, the more satisfied people are with public health services delivered by governments to patients, the higher the level of public QOL. From Model (1) to Model (3), there is a significantly positive relationship between public health service and QOL in terms of self-reported health, depressed mood, and happiness with correlation coefficients of 0.144, 0.167, and 0.328, respectively. In other words, the more satisfaction there was with government performance in providing public health service, the higher the degree of QOL. The more satisfaction with medical care and public health delivered by the government, the higher the level of QOL from Model (4) to (6). The correlation coefficients are 0.00921, 0.00905, and 0.0162, respectively.

Model (7) and Model (8) in Table 3 show that the relationship between expenditure on public health service and QOL is positively significant while controlling for sociodemographic and social capital variables, with its significance at 0.05 level. In other words, the more expenditure on public health, the higher the level of public QOL. However, the relationship between expenditure on public health and QOL is negative, with no significance suggested in Model (9) shown in Table 3. 

In addition, internet usage significantly increases the residents’ QOL, whereas the variation of QOL between males and females is quite unpredictable from the extant evidence. There seems to be a nonlinear relationship between age and QOL according to the models in Table 2. Years of education can enhance QOL in terms of depressed mood and happiness. Moreover, there is significant variation in QOL among different groups based on living place, party affiliation, and marriage status. Income level will significantly enhance residents’ QOL holding other variables constant in China.

### 3.3. Robust Check

The CGSS2015 applies the stratified sampling method, which might generate estimation bias without weighting. Therefore, this article reruns the regression models in order to get a robust parameter estimation by weighting. The results are shown in Table 4 and Table 5.

As shown in Table 4, the effects of performance in providing healthcare for the patients on QOL are significantly positive, with evidence in models (10), (11), and (12). Moreover, compared to the models without weighting, the coefficients are bigger than those in models (1), (2), and (3). Furthermore, the effects of satisfaction with medical care and public health delivered by the government on QOL are also bigger than those in models without weighting. The control variables are not explained here again. All told, the estimation of parameters should be weighted using data from CGSS2015. 

Models (16) and (17) show that when we estimate parameters based on weighting a multi-level ordered logit model, the expenditure on public heath can positively enhance public QOL in terms of self-reported health and depressed mood reduction, with a significance of 0.05. Although its effect is negative in happiness, the significance test has not passed.

The GDP per capita at province level can also help to reduce the public’s depression mood, holding other things constant. Besides, the effects of other control variables also changed after weighting methods are used in parameter estimation.

We also conducted a test to examine whether some important variables were omitted in our regression models. The link test and Ramsey test were used to prove that there were no significant omitting variable effects in our model using data CGSS2015. 

## 4. Discussion

Statistics show that the average level of QOL in China in 2015 was quite good. In this study, the effects of public health service performance on QOL in China was examined. Using subjective and objective methods to measure the performance of the public sector health system, this study clearly supported our research hypotheses that there are significant and positive effects of the performance of public-sector health systems on QOL, which are confirmed by various statistical methods. 

This research expanded upon previous work by linking the performance of the public sector health system with QOL. The results reveal that both subjective and objective measures for public health service show that its effect on QOL is robust and consistent. In addition, the weighting parameter estimation methods are also applied in this article to get robust results. Our research also investigated the new driving forces of QOL, the role of government health administration. Previous research mainly focused on the effect of social and economic factors, and events on QOL, while the role of government administration in public health was ignored. In this study, we used a weighted ordered logit model and a two-level hierarchical regression model to estimate the causal effect of public health service performance on QOL in China. 

In addition, based on the IPOO model, we also found evidence in public administration that there is a positive and robust relationship between the output indicator, public health expenditure, and the outcome variable on QOL. This paper provides evidence to link the two variables within the IPOO model in public sector health administration. 

What we have found has many implications for public health and QOL in China, and in other countries with similar institutional background and cultural traditions around the world. First, governments in China should shift their attention to public health investment instead of economic development. The economic development and disparity among urban and rural area have caused huge inequality in health service access in China. Therefore, the government should pay more attention to the equality of public health services across regions since it is a basic public service. For instance, the central government can diminish the inequality of public health services among provinces through a transfer of payment and other institutional arrangements. Second, more attention should be paid to the relationship between input-process-output and the outcome of the public health service. After four decades of reform and development, significant progress has been made in developing a system of medical and healthcare services. However, the development of public health is not only the mission of public agencies, but is also a complex system to build. Many factors such as human resources, funding, and institutional designs are supposed to be functionally arranged to gain high-quality management of public sector health regime. Third, QOL development should be linked with public health service enhancement in developing countries and it is an efficient tool for human rights. QOL is considered to be a barometer of a high quality of life in developed and developing countries, which can be improved by the reform and development of public sector health systems according to the evidence from our research. 

There are some limitations to the present study that should be addressed. Public health services can be a broad category and some policies or implementation processes can also influence the variation of QOL. This is difficult to explore with extant data. In addition, the cross-sectional data used in this paper could not support the causal relationship between public health services and QOL, which should be investigated in future research. With currently existing data, it is still difficult to quantify the relationship between the overall quality of life and health-related quality of life.

## 5. Conclusions

The government is increasingly playing an important role in public QOL in China. This paper attempted to examine the role of performance of public sector health systems measured by governments’ performance in providing healthcare for patients, public satisfaction with medical care and public health, and public expenditure on health. Our results conclude that a well-performing public sector health system has a significantly positive impact on QOL, indicated by self-reported health, depressed mood, and happiness. The subjective measurement of public health performance is a strong predictor of individuals’ QOL, however, the objective measurement of public health performance is not significant for the prediction of happiness. This research supports the application of IPOO in public sector health system evaluation by the public in terms of QOL. These findings call for policy action at a national and local level to enhance the degree of public health. For example, the government could implement higher expenditure on public health in order to increase people’s satisfaction with public health services in China. 

## Figures and Tables

**Figure 1 ijerph-17-02896-f001:**
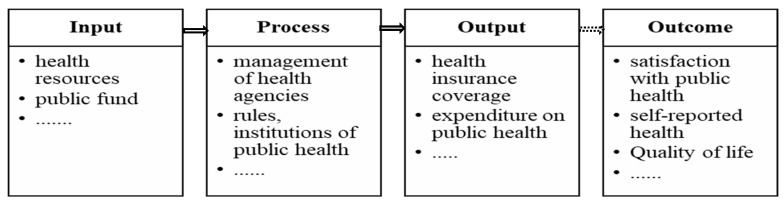
The Input-Process-Output-Outcome (IPOO) model for public health administration.

**Figure 2 ijerph-17-02896-f002:**
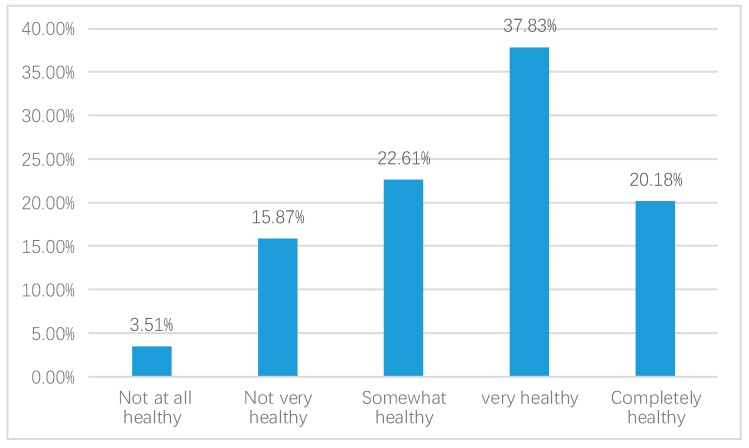
Self-reported health (weighted).

**Figure 3 ijerph-17-02896-f003:**
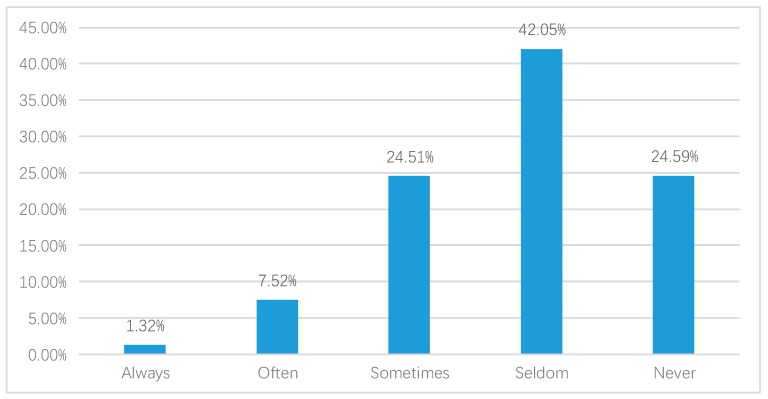
Depressed mood (weighted).

**Figure 4 ijerph-17-02896-f004:**
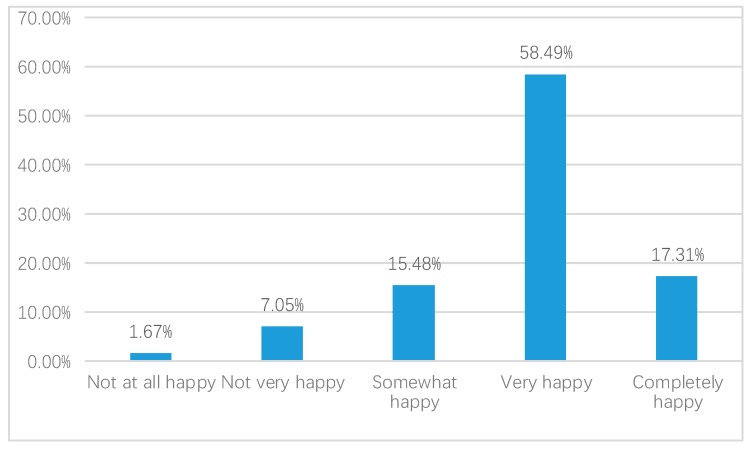
Happiness (weighted).

**Table 1 ijerph-17-02896-t001:** Variables description.

Variable	Observation	Mean	Std. Dev.	Min	Max
Self-reported health	10,961	3.608	1.075	1	5
Depressed mood	10,942	3.840	0.924	1	5
Happiness	10,953	3.867	0.821	1	5
HCP	10,846	3.365	0.910	1	5
MCPH	10,735	69.756	17.932	0	100
Internet usage	10,951	2.370	1.635	1	5
Gender					
Female	10,968	0.532	0.499	0	1
Age	10,968	50.397	16.898	18	95
Age squared	10,968	2825.388	1742.334	324	9025
Years of education	10,939	8.694	4.707	0	19
Place of residence					
Rural area	10,968	0.410	0.492	0	1
Party status					
Youth League member	10,921	0.050	0.218	0	1
Democratic party member	10,921	0.001	0.038	0	1
CPC member	10,921	0.104	0.305	0	1
Marital status					
Married	10,968	0.784	0.411	0	1
Divorced	10,968	0.021	0.143	0	1
Widowed	10,968	0.092	0.28	0	1
Income (log)	8722	9.765	1.271	3.912	16.118
Expenditure on public health (100 million)	28	348.104	169.954	65.27	777.55
GDP per capita (yuan RMB)	28	52,292.61	22,652.38	26,433	105,231
Population (10,000)	28	4740.25	2709.963	583	10,724

Note: HCP stands for the governments’ performance in providing healthcare for the patients; MCPH stands for satisfaction with medical care and public health.

**Table 2 ijerph-17-02896-t002:** Ordered logit model regression results.

Main	(1)	(2)	(3)	(4)	(5)	(6)
	Self-Reported Health	Depressed Mood	Happiness	Self-Reported Health	Depressed Mood	Happiness
HCP	0.144 ***	0.167 ***	0.328 ***			
	(4.30)	(3.98)	(8.73)			
MCPH				0.00921 ***	0.00905 ***	0.0162 ***
				(5.37)	(5.75)	(9.56)
Internet Usage	0.0702 ***	0.0765 ***	0.0436	0.0680 ***	0.0704 ***	0.0326
	(3.49)	(4.34)	(1.66)	(3.34)	(3.75)	(1.25)
Female	−0.205 ***	−0.0761	0.252 ***	−0.215 ***	−0.0779	0.243 ***
	(−3.61)	(−1.65)	(3.83)	(−3.88)	(−1.67)	(3.75)
Age	−0.0735 ***	−0.0222	−0.0583 ***	−0.0747 ***	−0.0244	−0.0595 ***
	(−7.28)	(−1.50)	(−5.29)	(−7.29)	(−1.73)	(−5.41)
Age Squared	0.000329 ***	0.000205	0.000668 ***	0.000336 ***	0.000228	0.000677 ***
	(3.70)	(1.57)	(6.59)	(3.77)	(1.84)	(6.74)
Years of Education	0.0115	0.0345 ***	0.0331 **	0.0120	0.0357 ***	0.0308 **
	(1.42)	(4.61)	(3.12)	(1.53)	(4.84)	(2.95)
Living Place	Reference group: urban					
Rural Area	0.0105	−0.0477	0.179	0.00107	−0.0503	0.179 *
	(0.16)	(−0.62)	(1.91)	(0.02)	(−0.66)	(1.97)
Party Affiliation	The mass as reference group					
Youth League	0.189	−0.0747	0.484 ***	0.176	−0.107	0.484 ***
	(1.26)	(−0.57)	(4.40)	(1.24)	(−0.81)	(4.37)
Democratic Member	0.359	0.323	0.0676	0.332	0.295	0.0562
	(1.19)	(0.46)	(0.08)	(1.06)	(0.42)	(0.07)
CPC Member	0.177 **	0.113	0.294 ***	0.172 **	0.0882	0.281 ***
	(3.04)	(1.44)	(3.87)	(2.91)	(1.09)	(3.86)
Marriage Status	Unmarried as reference group					
Married	0.302 ***	0.134	0.681 ***	0.295 ***	0.110	0.637 ***
	(3.57)	(1.30)	(5.45)	(3.55)	(1.02)	(5.03)
Divorced	−0.00505	−0.367 *	−0.329 *	0.0220	−0.400 *	−0.311 *
	(−0.04)	(−2.07)	(−2.26)	(0.16)	(−2.19)	(−2.30)
Widowed	0.323 **	0.0213	0.348 **	0.355 ***	0.0121	0.307 *
	(3.17)	(0.16)	(2.64)	(3.37)	(0.09)	(2.16)
Income per Year (ln)	0.175 ***	0.199 ***	0.177 ***	0.169 ***	0.199 ***	0.178 ***
	(4.78)	(6.52)	(4.09)	(4.68)	(6.38)	(4.34)
cut1	−4.035 ***	−2.196 ***	−1.526 **	−3.984 ***	−2.203 ***	−1.628 **
	(−9.42)	(−3.77)	(−2.84)	(−8.54)	(−3.88)	(−3.04)
cut2	−1.967 ***	−0.0836	0.308	−1.910 ***	−0.0877	0.222
	(−4.86)	(−0.16)	(0.57)	(−4.33)	(−0.17)	(0.42)
cut3	−0.637	1.662 ***	1.649 **	−0.577	1.665 **	1.558 **
	(−1.65)	(3.29)	(3.17)	(−1.38)	(3.27)	(2.98)
cut4	1.349 ***	3.617 ***	4.604 ***	1.413 **	3.615 ***	4.505 ***
	(3.34)	(7.33)	(8.42)	(3.25)	(7.34)	(8.17)
*N*	8561	8550	8559	8509	8503	8508

Note: t statistics in parentheses; * *p* < 0.05, ** *p* < 0.01, *** *p* < 0.001. HCP stands for the governments’ performance in providing healthcare for the patients; MCPH stands for the level of satisfaction with medical care and public health. The parallel assumption is not violated by Brant test.

**Table 3 ijerph-17-02896-t003:** Two-level hierarchical regression model.

Main	(7)	(8)	(9)
	Self-Reported Health	Depressed Mood	Happiness
Expenditure on Public Health (log)	0.236	0.245 *	−0.276
	(1.80)	(2.00)	(−1.94)
GDP Per Capita (log)	−0.108	0.403 *	0.0592
	(−0.62)	(2.50)	(0.32)
Internet Usage	0.0622 ***	0.0382 *	0.00977
	(3.32)	(2.00)	(0.48)
Female	−0.184 ***	−0.106 *	0.251 ***
	(−4.36)	(−2.50)	(5.52)
Age	−0.0701 ***	−0.0260 **	−0.0578 ***
	(−7.97)	(−2.95)	(−6.17)
Age Squared	0.000311 ***	0.000206 *	0.000680 ***
	(3.89)	(2.56)	(7.93)
Years of Education	0.0133 *	0.0276 ***	0.0285 ***
	(2.07)	(4.30)	(4.13)
Living Place	Reference group: urban		
Rural Area	−0.0421	0.0534	0.154 **
	(−0.82)	(1.05)	(2.80)
Party Affiliation	The mass as reference group		
Youth League	0.227 *	−0.0135	0.507 ***
	(1.84)	(−0.11)	(3.86)
Democratic Member	0.288	0.271	0.0706
	(0.61)	(0.53)	(0.13)
CPC Member	0.189 **	0.154 *	0.291 ***
	(2.84)	(2.28)	(4.09)
Marriage Status	Unmarried as reference group		
Married	0.224 *	0.190 *	0.607 ***
	(2.51)	(2.11)	(6.46)
Divorced	−0.0585	−0.382 **	−0.358 *
	(−0.37)	(−2.35)	(−2.16)
Widowed	0.263 *	0.0913	0.243 *
	(2.26)	(0.78)	(1.95)
Income per Year (ln)	0.207 ***	0.160 ***	0.179 ***
	(9.51)	(7.51)	(7.76)
cut1	−4.005 *	2.398	−3.780
	(−2.00)	(1.28)	(−1.74)
cut2	−1.931	4.498 *	−1.956
	(−0.96)	(2.40)	(−0.90)
cut3	−0.580	6.265 ***	−0.611
	(−0.29)	(3.35)	(−0.28)
cut4	1.451	8.241 ***	2.377
	(0.72)	(4.40)	(1.10)
Variance (province)			
Constant	0.115 ***	0.0977 **	0.135 ***
	(3.37)	(3.05)	(3.33)
*N*	8649	8639	8648

Note: t statistics in parentheses; * *p* < 0.1, ** *p* < 0.05, *** *p* < 0.01. The parallel assumption is not violated by Brant test.

**Table 4 ijerph-17-02896-t004:** Weighted ordered logit model regression.

Main	(10)	(11)	(12)	(13)	(14)	(15)
	Self-Reported Health	Depressed Mood	Happiness	Self-Reported Health	Depressed Mood	Happiness
HCP	0.160 ***	0.176 ***	0.363 ***			
	(5.89)	(6.32)	(11.96)			
MCPH				0.0100 ***	0.0102 ***	0.0169 ***
				(7.15)	(6.94)	(10.77)
Internet Usage	0.0604 **	0.0756 ***	0.0402	0.0583 **	0.0711 **	0.0267
	(2.91)	(3.35)	(1.72)	(2.78)	(3.15)	(1.13)
Female	−0.215 ***	−0.0861	0.249 ***	−0.235 ***	−0.0925	0.234 ***
	(−4.49)	(−1.78)	(4.67)	(−4.90)	(−1.91)	(4.38)
Age	−0.0796 ***	−0.0110	−0.0584 ***	−0.0804 ***	−0.0121	−0.0584 ***
	(−8.06)	(−1.07)	(−5.34)	(−8.02)	(−1.18)	(−5.31)
Age Squared	0.000376 ***	0.0000992	0.000658 ***	0.000379 ***	0.000109	0.000657 ***
	(4.17)	(1.06)	(6.56)	(4.13)	(1.16)	(6.52)
Years of Education	0.00958	0.0297 ***	0.0291 ***	0.0114	0.0313 ***	0.0281 ***
	(1.33)	(4.22)	(3.72)	(1.56)	(4.43)	(3.55)
Living Place	Reference group: urban					
Rural Area	−0.0281	−0.117 *	0.145 *	−0.0316	−0.120 *	0.156 *
	(−0.50)	(−2.13)	(2.39)	(−0.56)	(−2.16)	(2.55)
Party Affiliation	The mass as reference group					
Youth League	0.100	−0.0482	0.559 ***	0.0689	−0.0820	0.563 ***
	(0.66)	(−0.35)	(3.82)	(0.46)	(−0.62)	(3.86)
Democratic Member	0.822	0.555	0.823	0.761	0.506	0.706
	(1.62)	(0.77)	(1.18)	(1.46)	(0.69)	(1.11)
CPC member	0.145 *	0.112	0.327 ***	0.135	0.0852	0.308 ***
	(1.98)	(1.45)	(4.26)	(1.83)	(1.10)	(3.96)
Marriage Status	Unmarried as reference group					
Married	0.329 ***	0.140	0.819 ***	0.325 ***	0.127	0.767 ***
	(3.51)	(1.35)	(7.34)	(3.41)	(1.23)	(6.98)
Divorced	−0.0205	−0.391	−0.318	−0.00328	−0.435 *	−0.320
	(−0.12)	(−1.88)	(−1.68)	(−0.02)	(−2.12)	(−1.67)
Widowed	0.351 **	0.0722	0.398 **	0.394 **	0.0762	0.354 *
	(2.82)	(0.54)	(2.72)	(3.14)	(0.57)	(2.42)
Income per year (ln)	0.179 ***	0.195 ***	0.178 ***	0.171 ***	0.195 ***	0.181 ***
	(7.08)	(8.28)	(6.78)	(6.70)	(8.22)	(6.90)
cut1	−4.141 ***	−1.818 ***	−1.153 **	−4.115 ***	−1.750 ***	−1.278 **
	(−11.49)	(−4.71)	(−2.86)	(−11.23)	(−4.50)	(−3.14)
cut2	−2.089 ***	0.188	0.559	−2.040 ***	0.249	0.467
	(−5.91)	(0.52)	(1.43)	(−5.67)	(0.68)	(1.19)
cut3	−0.750 *	1.892 ***	1.871 ***	−0.694	1.968 ***	1.774 ***
	(−2.13)	(5.23)	(4.79)	(−1.94)	(5.40)	(4.51)
cut4	1.220 ***	3.831 ***	4.784 ***	1.280 ***	3.906 ***	4.676 ***
	(3.47)	(10.52)	(12.11)	(3.58)	(10.65)	(11.78)
*N*	8536	8526	8534	8485	8479	8484

Note: t statistics in parentheses; * *p* < 0.05, ** *p* < 0.01, *** *p* < 0.001. HCP stands for the governments’ performance in providing healthcare for the patients; MCPH stands for the level of satisfaction with medical care and public health. The parallel assumption is not violated by Brant test.

**Table 5 ijerph-17-02896-t005:** Multi-level ordered logit model with weighting.

Main	(16)	(17)	(18)
	Self-Reported Health	Depressed Mood	Happiness
Expenditure on Public Health (log)	0.211 *	0.188 *	−0.322
	(1.99)	(2.48)	(−1.16)
GDP Per Capita (log)	−0.108	0.370 *	0.0539
	(−0.66)	(2.20)	(0.29)
Internet usage	0.0534 **	0.0417	0.00669
	(2.60)	(1.67)	(0.18)
Gender	Male as reference		
Female	−0.192 ***	−0.120 *	0.259 ***
	(−3.39)	(−2.23)	(4.37)
Age	−0.0748 ***	−0.0147	−0.0573 ***
	(−5.73)	(−1.03)	(−5.84)
Age Squared	0.000351 **	0.000102	0.000672 ***
	(3.05)	(0.80)	(7.23)
Years of Education	0.0135	0.0230 **	0.0279 **
	(1.36)	(3.08)	(3.01)
Living Place	Reference group: urban		
Rural Area	−0.0567	0.0216	0.170 *
	(−1.04)	(0.35)	(2.46)
Party Affiliation	The mass as reference group		
Youth League	0.139	0.0151	0.584 ***
	(0.81)	(0.09)	(4.41)
Democratic Member	0.652	0.497	0.717
	(1.46)	(0.54)	(1.08)
CPC member	0.154	0.167 *	0.342 ***
	(1.91)	(2.04)	(3.36)
Marriage status	Unmarried as reference group		
Married	0.265 **	0.221	0.749 ***
	(2.84)	(1.86)	(6.06)
Divorced	−0.0322	−0.392 *	−0.316 *
	(−0.20)	(−1.98)	(−2.01)
Widowed	0.313 **	0.163	0.315 *
	(2.90)	(1.14)	(2.03)
Income per year (ln)	0.219 ***	0.161 ***	0.188 ***
	(8.13)	(6.57)	(5.47)
cut1	−4.148 *	2.172	−3.656
	(−2.11)	(1.08)	(−1.77)
cut2	−2.086	4.164 *	−1.957
	(−1.08)	(2.09)	(−0.92)
cut3	−0.728	5.891 **	−0.645
	(−0.39)	(2.95)	(−0.31)
cut4	1.278	7.848 ***	2.285
	(0.67)	(3.93)	(1.07)
Variance (Province)			
constant	0.110 ***	0.0993 **	0.125 ***
	(4.62)	(3.28)	(4.44)
*N*	8624	8615	8623

Note: t statistics in parentheses; * *p* < 0.05, ** *p* < 0.01, *** *p* < 0.001. HCP stands for the governments’ performance in providing healthcare for the patients; MCPH stands for the level of satisfaction with medical care and public health. The parallel assumption is not violated by Brant test.

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
