# Peer review of "The Effects of Performance of Public Sector Health System on Quality of Life in China: Evidence from the CGSS2015"

_ijerph, 2020, doi:10.3390/ijerph17082896_

Round 1

Reviewer 1 Report

I appreciate the efforts you made to improve your manuscript. My previous concerns are addressed and I think the revised product is significantly improved.

Author Response

Thank you very much. we really appreciate your professional reviewing work and your positive comments to our manuscript. 

Reviewer 2 Report

 This reviewer commends the authors for their study titled, “The Effects of Performance of Public Sector Health System on Quality of Life in China: Evidence from the CGSS2015.”

              The most important determinants of health-related quality of life as assessed by standard methods consistently reflect population differences in educational level, household income, employment status, marital status, chronic diseases, and disability.   The overall quality of life is determined by material wellbeing and health; political stability and security,  family and community life; job security;  gender equality, and political freedom. (https://www.economist.com/media/pdf/QUALITY_OF_LIFE.pdf).

            It is not clear how the authors are able to determine the status of the quality of life and its association with public health system performance without taking into consideration all these variables.  The authors have conflated public sector performance with the performance of the public health system; the latter can be used as part of the assessment for health related-quality of life, not overall quality of life.  It is. also difficult to quantify how much of  the variance of the overall quality of life is explained by health-related quality of life.

            The World Health Organization designed an instrument for measuring the quality of life:

(The WHOQOL-100  and the WHOQOL-BREF) recommended for use in research so that comparisons can be made across different communities, states,  and nations in the world.

The authors have used quality of life measure instruments, without any known validity and reliability;  thus impossible to generalize the conclusions and compare it with other research done using standard instruments.

            I do not believe the variables used by the authors are standard and adequate to measure the health-related quality of life.  This study requires a total redesign, the use of appropriate instruments, a revised data analysis, and a total re-write-up.

Reviewer 3 Report

Comments are embedded within the PDF of your manuscript.  Overall, these are just minor editorial comments or requests for clarification on one or two points.  This is a good manuscript - good study and applicable information to both scientific community and political community.  

Round 2

Reviewer 2 Report

This reviewer commends the authors for putting together their study on the quality of life in China. The authors have clarified their study purpose and scope, verified the validity and reliability of survey instruments,  and acknowledged and included in the limitations section the relationship.  pertaining to the overall quality of life and health-related quality of life.

Author Response

Dear Reviewer,

Thank you very much for your professional reviewing, and accepting your suggesions will definitely improve our manuscript a lot. In this round of revision, we revise the manuscript in terms of the English language and abstract, which can be retrieved from the revised one.

Best regards,

This manuscript is a resubmission of an earlier submission. The following is a list of the peer review reports and author responses from that submission.

Round 1

Reviewer 1 Report

Thank you for the opportunity to review your manuscript.

P.2, lines 45-54: The authors make substantial claims necessary to support the foundation of their study yet provided no citations in support of those claims. This needs to be addressed.

P.3, lines 108-112: The differences in external barriers between developed/urban and lesser developed/rural areas may need to be addressed here. A single variable such as transportation (i.e. access to clinical care) could be a confounder that in part explains the significant differences in clinical visits. 

Discussion section, line 273: What are those implications?

Reviewer 2 Report

It would benefit the reader to know what CCGS means from the onset. Thus, please describe it in the abstract. 

I think a large current short coming of the study is non-communication of how the population's reported QOL connects with public health service quality. A separate survey on sentiments/ratings of public health services rendered would facilitate this connection. 

Also, explain what the 0.144, 0.167, and 0.328 mean. Quality of life should not be expressed as a digit but should be expressed as a degree of agreement. For example, "Relative to..., 23% of the population communicated..." and, to further assist you, use wording such as, "correlation coefficient of 0.144". 

Work would be timely but heavy lifting on editing and connecting the dots is still needed. 
